# Ensemble-in-One: Learning Ensemble within Random Gated Networks for Enhanced Adversarial Robustness

## Abstract

Adversarial attacks have threatened modern deep learning systems by crafting adversarial examples with small perturbations to fool the convolutional neural networks (CNNs). Ensemble training methods are promising to facilitate better adversarial robustness by diversifying the vulnerabilities among the sub-models, simultaneously maintaining comparable accuracy as standard training. Previous practices also demonstrate that enlarging the ensemble can improve the robustness. However, existing ensemble methods are with poor scalability, owing to the rapid complexity increase when including more sub-models in the ensemble. Moreover, it is usually infeasible to train or deploy an ensemble with substantial sub-models, owing to the tight hardware resource budget and latency requirement. In this work, we propose *Ensemble-in-One* (EIO), a simple but effective method to enlarge the ensemble within a random gated network (RGN). EIO augments the original model by replacing the parameterized layers with multi-path random gated blocks (RGBs) to construct an RGN. By diversifying the vulnerability of the numerous paths through the super-net, it provides high scalability because the paths within an RGN exponentially increase with the network depth. Our experiments demonstrate that EIO consistently outperforms previous ensemble training methods with even less computational overhead, simultaneously achieving better accuracy-robustness trade-offs than adversarial training.

## 1 Introduction

With the convolutional neural networks (CNNs) becoming ubiquitous, the security and robustness of neural networks is attracting increasing interests. Recent studies find that CNN models are inherently vulnerable to adversarial attacks (Goodfellow et al., 2014), which craft imperceptible perturbations on the images, referred to as adversarial examples, to mislead the neural network models. Even without accessing the target model, an adversary can still generate adversarial examples from other surrogate models to attack the target model by exploiting the adversarial transferability among them.

Such vulnerability of CNN models has spurred extensive researches on improving the robustness against adversarial attacks. One stream of approaches targets on learning robust features for an individual model (Madry et al., 2017; Brendel et al., 2020). Informally, robust features are defined as the features that are less sensitive to the adversarial perturbations added on the inputs. A representative approach, referred to as adversarial training (Madry et al., 2017), on-line generates adversarial examples on which the model minimizes the training loss. As a result, adversarial training encourages the model to learn the features that are less sensitive to the adversarial input perturbations, thereby alleviating the model's vulnerability. However, such adversarial training methods often have to sacrifice the clean accuracy for enhanced robustness (Zhang et al., 2019), since they exclude the non-robust features and become less distinguishable for the examples with high similarity in the feature space.

Besides empowering improved robustness for an individual model, another stream of researches focuses on forming strong *ensembles* to improve the robustness (Yang et al., 2020; Bagnall et al., 2017; Pang et al., 2019; Kariyappa & Qureshi, 2019). Generally speaking, an ensemble is constructed by aggregating multiple sub-models. Intuitively, an ensemble is promising to facilitate better robustness than an individual model because a successful attack needs to mislead the majority of the

| #sub-model | Baseline | ADP | GAL | DVERGE |
|:---:|:---:|:---:|:---:|:---:|
| 3 | 0.0%/1.5% | 0.0%/9.6% | 39.7%/11.4% | 53.2%/40.0% |
| 5 | 0.0%/2.1% | 0.0%/11.8% | 32.4%/31.7% | 57.2%/48.9% |
| 8 | 0.0%/3.2% | 0.0%/12.0% | 22.4%/37.0% | 63.6%/57.9% |

Table 1: Adversarial accuracy of the ensembles trained by different methods, with 3, 5, and 8 sub-models respectively (Yang et al., 2020). The number before and after the slash represent black-box and white-box adversarial accuracy under perturbation strength 0.03 and 0.01 respectively.

sub-models rather than just one. While the robustness of an ensemble highly relies on the diversity of the sub-models, recent study finds that CNN models trained independently on the same dataset are with highly-overlapped adversarial subspaces (Tramèr et al., 2017). Therefore, many studies propose ensemble training methods to diversify the sub-models. For example, DVERGE (Yang et al., 2020) proposes to distill non-robust features corresponding to each sub-model's vulnerability, then isolates the vulnerabilities of the sub-models by mutual learning such that impeding the adversarial transferability among them.

There is another learned insight that the ensembles composed by more sub-models tend to capture greater robustness improvement. Table 1 shows the robustness trend of the ensembles trained with various ensemble training methods. Robustness improvement can be obtained by including more sub-models within the ensemble. This drives us to further explore whether the trend will continue when keeping enlarging the ensemble. However, existing ensemble construction methods are with poor scalability because of the rapidly increasing overhead, especially with mutual learning which trains the sub-models in a round-robin manner, the complexity will rise at a speed of $O(n^2)$.

We propose *Ensemble-in-One*, a novel approach that can improve the scalability of ensemble training and introduce randomness mechanism for enhanced generalization, simultaneously obtaining better robustness and higher efficiency. For a dedicated CNN model, we conduct a *Random Gated Network* (RGN) by substituting each parameterized layer with a *Random Gated Block* (RGB) on top of the neural architecture. Through this, the network can instantiate numerous sub-models by controlling the gates in each block. Ensemble-in-One substantially reduces the complexity when scaling up the ensemble. In summary, the contributions of this work are listed as below:

- Ensemble-in-One is a simple but effective method that learns adversarially robust ensembles within one over-parametrized random gated network. The EIO construction enables us to employ ensemble learning techniques to learn more robust individual models with minimal computational overheads and no extra inference overhead.
- Extensive experiments demonstrate the effectiveness of Ensemble-in-One. It outperforms the previous ensemble training methods with even less computational overhead. Moreover, EIO also achieves better accuracy-robustness trade-offs than adversarial training method.

## 2 RELATED WORK

### 2.1 ADVERSARIAL ATTACKS AND COUNTERMEASURES.

The inherent vulnerability of CNN models poses challenges on the security of deep learning systems. An adversary can apply an additive perturbation on an original input to generate an adversarial example that induces wrong prediction in CNN models (Goodfellow et al., 2014). Denoting an input as $x$, the goal of adversarial attacks is to find a perturbation $\delta$ s.t. $x_{adv} = x + \delta$ can mislead the model, where $||\delta||_p$ satisfies the intensity constraint $||\delta||_p \leq \epsilon$. To formulate that, the adversarial attack aims at maximizing the loss $\mathcal{L}$ for the model with parameters $\theta$ on the input-label pair $(x, y)$, i.e. $\delta = \text{argmax}_\delta \mathcal{L}_\theta(x + \delta, y)$, under the constraint that the $\ell_p$ norm of the perturbation should not exceed the bound $\epsilon$. Usually, we use $\ell_\infty$ norm (Goodfellow et al., 2014; Madry et al., 2017) of the perturbations to measure the attack's effectiveness or model's robustness. An attack that requires smaller perturbation to successfully deceive the model is regarded to be stronger. Correspondingly, a defense that enforces the attacks to enlarge perturbation intensity is regarded to be more robust.

Various adversarial attack methods have been investigated to strengthen the attack effectiveness. The fast gradient sign method (FGSM) (Goodfellow et al., 2014) exploits the gradient descent method

to generate adversarial examples. As an improvement, many studies further show the attack can be strengthened through multi-step projected gradient descent (PGD) (Madry et al., 2017) generation, random-starting strategy, and momentum mechanism (Dong et al., 2017). Then SGM (Wu et al., 2020) further finds that adding weight to the gradients going through the skip connections can make the attacks more effective. Other prevalent attack approaches include C&W losses (Carlini & Wagner, 2017b) , M-DI$^2$-FGSM (Xie et al., 2019), etc. These attacks provide strong and effective ways to generate adversarial examples, rendering a huge threat to real-world deep learning systems.

To improve the robustness of CNN systems, there are also extensive countermeasures for adversarial attacks. One active research direction targets improving the robustness of individual models. Adversarial training (Madry et al., 2017) optimizes the model on the adversarial examples generated in every step of the training stage. Therefore, the optimized model will tend to drop non-robust features to converge better on the adversarial data. However, adversarial training encourages the model to fit the adversarial examples, thereby reducing the generalization on the clean data and causing significant degradation of the clean accuracy.

## 2.2 TEST-TIME RANDOMNESS FOR ADVERSARIAL DEFENSE

Besides the aforementioned training techniques, there exist studies that introduce test-time randomness to improve the robustness. Feinman et. al. (Feinman et al., 2017) utilize the uncertainty measure in dropout networks to detect adversarial examples. Dhillon et. al. (Dhillon et al., 2018) and Xie et. al. (Xie et al., 2017) incorporate layer-wise weighted dropout and random input transformations during test time to improve the robustness. Test-time randomness is found to be effective in increasing the required distortion on the model, since test-time randomness makes generating white-box adversarial examples almost as difficult as generating transferable black-box ones (Carlini & Wagner, 2017a). Nevertheless, test-time randomness increases the inference cost and can be circumvented to some extent with the expectation-over-transformation technique (Athalye et al., 2018).

## 2.3 ENSEMBLE TRAINING FOR ADVERSARIAL DEFENSE.

Besides improving the robustness of individual models, another recent research direction is to investigate the robustness of model ensembles in which multiple sub-models work together. The basic idea is that multiple sub-models can provide diverse decisions. Ensemble methods can combine multiple weak models to jointly make decisions, thereby assembling as a stronger entirety. However, it is demonstrated that independent training of multiple models tends to capture similar features, which would not provide diversities among them (Kariyappa & Qureshi, 2019).

Therefore, several studies propose ensemble training methods to fully diversify the sub-models to improve the ensemble robustness. For example, Pan et. al. treat the distribution of output predictions as a diversity measurement and they propose an adaptive diversity promoting (ADP) regularizer (Pang et al., 2019) to diversify the non-max predictions of sub-models. Sanjay et. al. regard the gradients w.r.t. the inputs as a discrimination of different models, thus they propose a gradient alignment loss (GAL) (Kariyappa & Qureshi, 2019) which takes the cosine similarity of the gradients as a criterion to train the sub-models. The very recent work DVERGE (Yang et al., 2020) claims that the similar non-robust features captured by the sub-models cause high adversarial transferability among them. Therefore, the authors exploit non-robust feature distillation and adopt mutual learning to diversify and isolate the vulnerabilities among the sub-models, such that the within-ensemble transferability is highly impeded. However, as mentioned before, such ensemble methods are overwhelmed by the fast-increasing overhead when scaling up the ensemble. For example, DVERGE takes 11 hours to train an ensemble with three sub-models while needs approximately 50 hours when the sub-model count increases to eight. Therefore, a more efficient ensemble construction method is highly demanded to tackle the scaling problem.

## 3 ENSEMBLE-IN-ONE

### 3.1 BASIC MOTIVATION

The conventional way to construct ensembles is to simply aggregate multiple sub-models by averaging their predictions, which is inefficient and hard to scale up. An intuitive way to enhance the

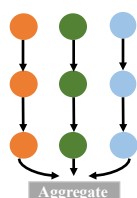 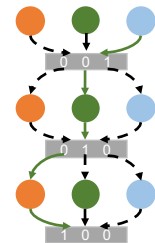

Figure 1: Normal ensemble training of multiple sub-models (Left) and the proposed ensemble-in-one training within a random gated network (Right). By selecting the paths along augmented layers, the ensemble-in-one network can instantiate $n^L$ sub-models, where $n$ represents the augmentation factor of the multi-gated block for each augmented layer and $L$ represents the number of augmented layers in the network.

scalability of the ensemble construction is to introduce an ensemble for each layer in the network. As shown in Fig.1, we can augment a dynamic network by augmenting each parameterized layer with an $n$-path gated block. Then by selecting the paths along the augmented layer, the dynamic network can instantiate $n^L$ varied sub-models ideally. Taking ResNet-20 as an example, by replacing each convolution layer (ignoring the skip connection branch) with a two-path gated module, the overall path count will approach $2^{19} = 524288$. Such augmentation way provides an approximation to training a very large ensemble of sub-models. Then through vulnerability diversification mutual learning, each path tends to capture better robustness. Following this idea, we propose *Ensemble-in-One* to further improve the robustness of both individual models and ensembles.

## 3.2 CONSTRUCTION OF THE RANDOM GATED NETWORK

Denote a candidate neural network as $\mathcal{N}(o_1, o_2, ..., o_m)$, where $o_i$ represents an operator in the network. To transform the original network into a random gated network (RGN), we first extract the neural architecture to obtain the connection topology and layer types. On top of that, we replace each parameterized layer (mainly convolutional layer, optionally followed by a batch normalization layer) with a random gated block (RGB). As shown in Fig. 2, each RGB simply repeats the original layer by $n$ times, and leverages binary gates with uniform probabilities to control the open or mutation of corresponding sub-layers. These repeated sub-layers are with different weight parameters. We denote the RGN as $\mathcal{N}(d_1, d_2, ..., d_m)$, where $d_i = (o_{i1}, ..., o_{in})$. Let $g_i$ be the gate information in the $i_{\text{th}}$ RGB, then a specific path derived from the RGN can be expressed as $\mathcal{P} = (g_1 \cdot d_1, g_2 \cdot d_2, ..., g_m \cdot d_m)$.

For each RGB, when performing the computation, only one of the $n$ gates is opened at a time, and the others will be temporarily muted. Thus by, only one path of activation is active in memory during training, which reduces the memory occupation of training an RGN to the same level of training an individual model. Moreover, to ensure that all paths can be equally sampled and trained, each gate in a RGB is chosen with identical probability, i.e. $1/n$ if each RGB consists of $n$ sub-operators. Therefore, the binary gate function can be expressed as:

$$g_i = \begin{cases} [1, 0, ..., 0] & \text{with probability } 1/n, \\ [0, 1, ..., 0] & \text{with probability } 1/n, \\ \quad ... \\ [0, 0, ..., 1] & \text{with probability } 1/n. \end{cases} \tag{1}$$

An RGN is analogous to the super network in parameter-sharing neural architecture search, and the forward process of an RGN is similar to evaluating a sub-architecture (Pham et al., 2018; Cai et al., 2018). Compared to conventional ensemble training methods, our method is easier to scale up the ensemble. It only incurs $n\times$ memory occupation for the weight storage, while still keeping the same memory requirement for activation as an individual model.

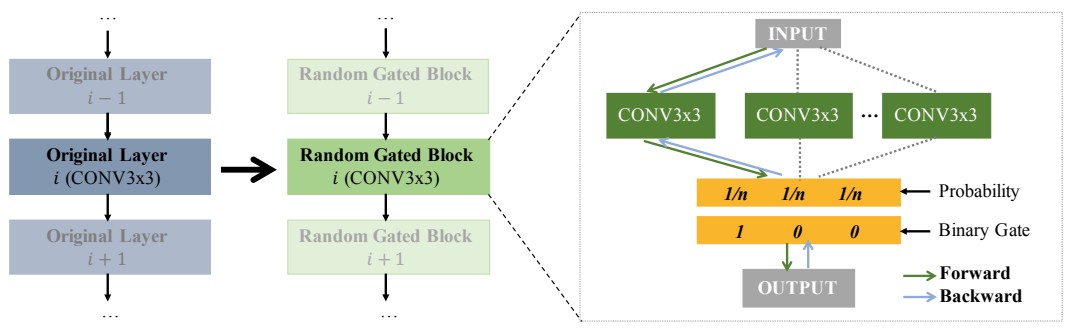

(a) Construction of random gated network    (b) The schematic diagram of random gated block

Figure 2: The construction of random gated network based on random gated blocks. The forward propagation will select one path to allow the input pass. Correspondingly, the gradients will also propagate backward along the same path.

### 3.3 LEARNING ENSEMBLE IN ONE

The goal of learning ensemble-in-one is to encourage the vulnerabilities diversity of all the paths within the RGN by mutually learning from each other. Let $\mathcal{P}_i$ and $\mathcal{P}_j$ be two different paths, where we define two paths as different when at least one of their gates is different. To diversify the vulnerabilities, we need first distill the non-robust features of the paths so that the optimization process can isolate them. We adopt the same non-robust feature distillation strategy as previous work (Ilyas et al., 2019; Yang et al., 2020). Consider two randomly-sampled independent input-label pairs $(x_t, y_t)$ and $(x_s, y_s)$ from the training dataset, the distilled feature of $x_t$ corresponding to $x_s$ by the $l_{\text{th}}$ layer of path $\mathcal{P}_i$ can be achieved by:

$$x'_{\mathcal{P}_i^l}(x_t, x_s) = \text{argmin}_z ||f_{\mathcal{P}_i}^l(z) - f_{\mathcal{P}_i}^l(x_t)||^2, \quad s.t. \quad ||z - x_s||_\infty \le \epsilon_d. \tag{2}$$

Such feature distillation aims to construct a sample $x'_{\mathcal{P}_i^l}$ by adding perturbations on $x_s$ so that the response in $l_{\text{th}}$ layer of $\mathcal{P}_i$ of $x'_{\mathcal{P}_i^l}$ is similar as that of $x_t$, while the two inputs $x_t$ and $x_s$ are completely different and independent. This exposes the vulnerability of path $\mathcal{P}_i$ on classifying $x_s$. Therefore, for another different path $\mathcal{P}_j$, it can learn on the distilled data to correctly classify them to circumvent the vulnerability. The optimization objective for path $\mathcal{P}_j$ is to minimize:

$$\mathbb{E}_{(x_t, y_t), (x_s, y_s), l} \mathcal{L}_{f_{\mathcal{P}_j}}(x'_{\mathcal{P}_i^l}(x_t, x_s), y_s). \tag{3}$$

As it is desired that each path can learn from the vulnerabilities of all the other paths, the objective of training the ensemble-in-one RGN is to minimize:

$$\sum_{\forall \mathcal{P}_j \in \mathcal{N}} \mathbb{E}_{(x_t, y_t), (x_s, y_s), l} \sum_{\forall \mathcal{P}_i \in \mathcal{N}, i \ne j} \mathcal{L}_{f_{\mathcal{P}_j}}(x'_{\mathcal{P}_i^l}(x_t, x_s), y_s), \tag{4}$$

where $\mathcal{N}$ is the set of all paths in the RGN. While it is obviously impossible to involve all the paths in a training iteration, we randomly sample a certain number of paths by stochastically set the binary gates according to Eq.1. We denote the number of paths sampled in each iteration as $p$. Then the selected paths can temporarily combine as a subset of the RGN, referred to as $\mathcal{S}$. The paths in the set $\mathcal{S}$ keep changing throughout the whole training process, such that all paths will have equal opportunities to be trained.

The training process of the RGN is summarized by the pseudo-code in Algorithm 1. Before starting vulnerability diversification training, we pre-train the RGN based on standard training settings to help the RGN obtain basic capabilities. The process is simple, where a random path will be sampled in each iteration and trained on clean data. Then for each batched data, the process of vulnerability diversification contains three basic steps. First, randomly sample $p$ paths to be involved in the iteration. Note that the sampled paths should be varied, i.e. if the distilling layer is set to $l$, for any $\mathcal{P}_i$, $\mathcal{P}_j$ in $\mathcal{S}$, there must be at least one different gate among the top $l$ gates, i.e. $\exists k \in [1, l]$, s.t. $\mathcal{P}_i[k] \ne \mathcal{P}_j[k]$. Second, distilling the vulnerable features of the sampled paths according to

---

**Algorithm 1** Training process for learning Ensemble-in-One

---

**Require:** Path samples per ietration $p$
**Require:** Random Gated Network $\mathcal{N}$ with $L$ parameterized layers
**Require:** Pre-training epoch $E_w$, training epoch $E$, and data batch $B_d$
**Require:** Optimization loss $\mathcal{L}$, learning rate $lr$
**Ensure:** Trained Ensemble-in-One model
1: # pre-training of $\mathcal{N}$
2: **for** e = 1, 2, ..., $E_w$ **do**
3:     **for** b = 1, 2, ..., $B_d$ **do**
4:         Random Sample Path $\mathcal{P}_i$ from $\mathcal{N}$
5:         Train $\mathcal{P}_i$ in batched data
6:     **end for**
7: **end for**
8: # learning vulnerability diversity for $\mathcal{N}$
9: **for** e = 1, 2, ..., $E$) **do**
10:     **for** b = 1, 2, ..., $B_d$) **do**
11:         Random sample $l \in [1, L]$
12:         # randomly sample $p$ paths
13:         $\mathcal{S}$=[$\mathcal{P}_1, \mathcal{P}_2, ..., \mathcal{P}_p$], s.t. $\forall i, j, \exists k \in [1, l]$, s.t. $\mathcal{P}_i[k] \neq \mathcal{P}_j[k]$
14:         Get data $(X_t, Y_t), (X_s, Y_s) \leftarrow D$
15:         # Get distilled data
16:         **for** i = 1, 2, ..., $p$ **do**
17:             $X'_i = x'_{\mathcal{P}^l_i}(X_t, X_s)$
18:         **end for**
19:         $\nabla_{\mathcal{N}} \leftarrow 0$
20:         **for** i = 1, 2, ..., $p$ **do**
21:             $\nabla_{\mathcal{P}_i} = \nabla(\sum_{j \neq i} \mathcal{L}_{f_{\mathcal{P}_i}}(f_{\mathcal{P}_i}(X'_j), Y_s))$
22:             $\nabla_{\mathcal{N}} = \nabla_{\mathcal{N}} + \nabla_{\mathcal{P}_i}$
23:         **end for**
24:         $\mathcal{N} = \mathcal{N} - lr * \nabla_{\mathcal{N}}$
25:     **end for**
26: **end for**

---

Eq. 2. The distillation process is the same as proposed in DVERGE, by applying a PGD scheme for approximating the optimal perturbations. Third, mutually train each path with the distilled data from the other paths in a round-robin manner. Because the paths unavoidably share a proportion of weights owing to the weight sharing mechanism in super-net, the gradients of the weights will not be updated until all sampled paths are included.

## 3.4 MODEL DERIVATION AND DEPLOYMENT

Once the training of RGN is finished, we can then derive and deploy the model in two ways. One way is to deploy the entire RGN, then in inference stage, the gates throughout the network will be randomly selected to process an input. The advantage is that the computation is randomized, which may beneficial for improving the robustness under white-box attacks, because the transferability among different paths was impeded during diversity training. However, the disadvantage is that the accuracy is unstable owing to the dynamic choice of inference path, where the fluctuation reaches 1-2 percentage.

Another way is to derive individual models from the RGN. By sampling a random path and eliminating the other redundant modules, an individual model can be rolled out. We can also sample multiple paths and derive multiple models to combine as an ensemble. Deploying models in this way ensures the stability of the prediction as the randomness is eliminated. In addition, the derived models can be slightly finetuned with small learning rate for a few epochs to compensate for the under-convergence, as the training process of RGN cannot fully train all paths as the probability of each specific path being sampled is relatively low. In our implementation, we exploit the latter method to derive an individual model for deployment.

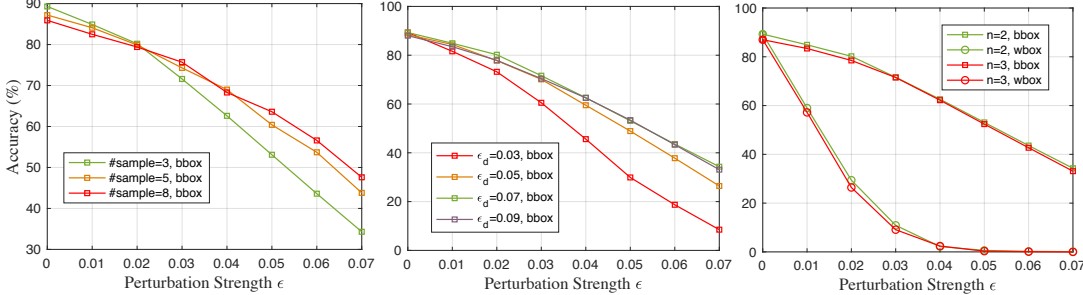

Figure 3: Investigation on the hyper-parameters involved in the Ensemble-in-One construction and training. All these experiments are implemented on ResNet-20 over CIFAR-10 dataset. Left: The black-box adversarial accuracy under different sample count $p$ per iteration; Middle: The black-box adversarial accuracy under different distillation perturbation $\epsilon_d$; and Right: the adversarial accuracy under different augmentation factor $n$.

## 4 EXPERIMENTAL RESULTS

### 4.1 EXPERIMENT SETTINGS

**Benchmark.** The experiments are constructed on the ResNet-20 (He et al., 2016) and VGG-16 networks with the CIFAR dataset (Krizhevsky et al., 2009). Specifically, we construct the ResNet-20 and VGG-16 based RGNs by substituting each convolution layer to a $n$-path RGB (in default $n = 2$). Overall, there are 19 RGBs (containing 19 convolution layers in the straight-through branch) for ResNet-20 and 14 RGBs for VGG-16 (containing the 14 convolution layers). To evaluate the effectiveness of our method, we compare Ensemble-in-One with multiple counterparts, including the *Baseline* which trains the models in a standard way and three previous ensemble training methods: *ADP* (Pang et al., 2019), *GAL* (Kariyappa & Qureshi, 2019), and *DVERGE* (Yang et al., 2020). Meanwhile, we also add the adversarial training (*AdvT*) method into the comparison.

**Training Details.** The trained ensemble models of Baseline, ADP, GAL, and DVERGE are referred to the implementation which is publicly released in (Yang et al., 2020) [1]. We train the Ensemble-in-One networks for 200 epochs using SGD with momentum 0.9 and weight decay 0.0001. The initial learning rate is 0.1, and decayed by 10x at the 100-th and the 150-th epochs respectively. When deriving the individual models, we fine-tune the derived models for 0-20 epochs using SGD with learning rate 0.001, momentum 0.9 and weight decay 0.0001. Note that the fine-tuning process is optional and can adjust the epochs for a dedicated model. In default, for an RGN training, we sample 3 paths to construct temporary sub-ensemble per iteration. The augmentation factor $n$ for each RGB is set to 2, and the PGD-based perturbation strength $\epsilon_d$ for feature distillation is set to 0.07 with 10 iterative steps and each step size of $\epsilon_d/10$.

**Attack Models.** We categorize the adversarial attacks as black-box transfer attacks and white-box attacks. The white-box attack assumes the adversary has full knowledge of the model parameters and architectures, and the black-box attack assumes the adversary cannot access the target model and can only generate adversarial examples from surrogate models to launch the attacks. For fair comparison, we adopt exactly the same attack methodologies and the same surrogate models as DVERGE to evaluate the robustness. The detailed adversarial settings can be found in Appendix A.1. We believe the attacks are powerful and can identify the robustness of the various models.

### 4.2 ROBUSTNESS EVALUATION

**Hyper-parameter Exploration.** Recall that three important hyper-parameters are involved in the training procedure. One is the count of sampled paths $p$ to participate in each training iteration, one is the strength of feature distillation perturbation $\epsilon_d$ as illustrated in Eq.2, and the other is the augmentation factor $n$ for constructing the RGN, i.e. how many times will an operator be repeated

---

[1]https://github.com/zjysteven/DVERGE

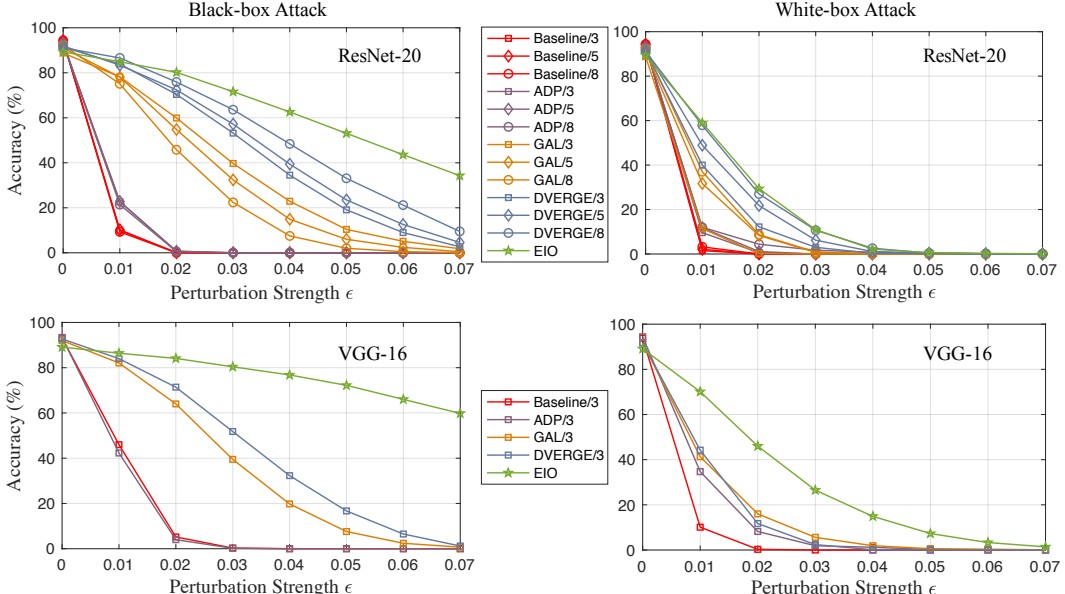

Figure 4: Contrasting the robustness of Ensemble-in-One with previous ensemble training methods. Left: adversarial accuracy under black-box transfer attack; and right: adversarial accuracy under white-box attack. The number after the slash stands for the number of sub-models within the ensemble. The evaluations include ResNet-20 and VGG-16 over the CIFAR-10 dataset. The distillation perturbation strength of VGG-16-based EIO is set as $\epsilon_d = 0.03$.

to build a RGB. We make experiments to investigate the impact of these hyper-parameters on the clean accuracy and the adversarial robustness.

Fig.3 (Left) shows the curves of black-box adversarial accuracy under different sampled path count $p$ per training iteration. As is observed, when the sampled paths increase, the robustness of the derived individual model also improves. The underlying reason is that more samples of paths participating in each iteration allows more paths to be mutually trained, thereby each path is expected to learn from more diverse vulnerabilities. However, the clean accuracy drops with the increasing of path sample count, because a single operator has to adapt to diverse paths simultaneously. Moreover, the training time will also increase as the training complexity satisfies $\mathcal{O}(p^2)$.

Fig.3 (Middle) shows the curves of black-box adversarial accuracy under different feature distillation $\epsilon_d$. We find similar conclusions as presented in DVERGE. A larger $\epsilon_d$ can push the distilled data $x'_{\mathcal{P}_i^l}(x_t, x_s)$ share more similar internal representation as $x_t$. While the objective is to reduce the loss of $\mathcal{P}_j$ on classifying $x'_{\mathcal{P}_i^l}$, the larger loss will boost the effectiveness of learning the vulnerability, thereby achieving better robustness. However, we also find the clean accuracy drops with the increase of $\epsilon_d$. And there exists a switching point where it will stop obtaining robustness improvement from continually increasing $\epsilon_d$. The experimental results suggest $\epsilon_d = 0.07$ to achieve higher robustness and clean accuracy simultaneously.

Fig.3 (Right) shows the comparison of adversarial accuracy when applying different augmentation factor $n$ for constructing the RGN. Observe that increasing the factor $n$ brings no benefit on either the clean accuracy or adversarial accuracy. It stands to reason that augmenting $2\times$ operators for each RGB has already provided sufficient candidate paths. Moreover, increasing the $n$ leads to more severe under-convergence of training because each path has a decreased probability of being sampled. Therefore, we suggest the augmentation factor as $n=2$ for each convolution layer.

**Comparison with Ensemble Methods.** Fig.4 shows the overall adversarial accuracy of the models trained by different methods with a wide range of attack perturbation strength. ResNet-20 and VGG-16 are selected as the basic network to construct the ensembles and the EIO super-networks. The results show that through our Ensemble-in-One method, an individual model derived from the RGN can significantly outperform the heavy ensembles trained by previous ensemble training meth-

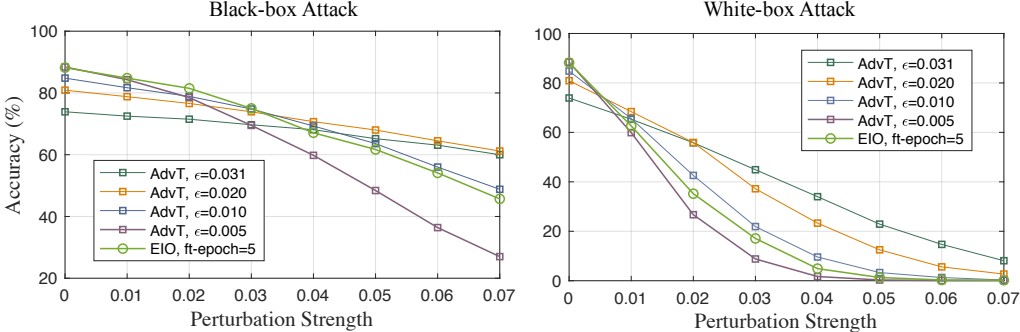

Figure 5: Contrasting the robustness of Ensemble-in-One and AdvT with different adversarial perturbation settings. The experiments are implemented on ResNet-20 over CIFAR-10. The "ft-epoch" means the fine-tuning epoch of the derived model. When aligning the clean accuracy, EIO achieves better robustness than AdvT.

ods with higher adversarial accuracy under both black-box and white-box attacks, simultaneously achieving comparable clean accuracy. More experiments and analysis can refer to the Appendix A.2, A.3, and A.4. These results demonstrate that we successfully train an ensemble within one RGN network and improves the robustness of an individual model to outperform the ensembles such that the deployment overhead can be substantially reduced.

**Comparison with Adversarial Training.** AdvT has been demonstrated as a promising approach on enhancing the robustness. Prior work attributes the enhancement to the exclusion of non-robust features during AdvT. However, these non-robust features might be *useful* to the classification accuracy, resulting in trade-offs between the clean accuracy and the robustness. One can adjust the perturbation strength in the AdvT to acquire different combinations of clean accuracy and adversarial robustness, as shown in Fig.5. It can be figured out that EIO significantly outperforms AdvT when aligning their clean accuracy (AdvT w/ $\epsilon = 0.005$), which suggests that EIO learns more *useful, robust* features while excluding more *useless, non-robust* features than AdvT.

## 5    DISCUSSION & FUTURE WORK

There are also several points that are worthy further exploration while we leave to future work. First, current implementation of augmenting the RGN is simple, by repeating the convolution layers for multiple times. Nevertheless, as observed in Fig.3 (Right), enlarging the augmentation factor brings no benefit on improving the robustness. Hence, there might be better ways of constructing RGNs that can compose stronger randomized network, e.g. subtracting some of the unnecessary RGBs or augmenting by diverse operators instead of simply repeating. Second, although black-box attacks are more prevalent in real world, defending against white-box attacks is still in demand because recent research warns the high risks of exposing the private models to the adversary (Hua et al., 2018; Hu et al., 2020). Randomized multi-path network can provide promising solutions to alleviating the white-box threats. If the adversarial transferability among the different paths can be impeded, the adversarial example generated from one path will be ineffective for another path. Hence, it will make the white-box attacks as difficult as black-box transfer attacks. We believe it is a valuable direction to explore defensive method based on randomized multi-path network.

## 6    CONCLUSIONS

In this work, we propose Ensemble-in-One, a novel approach that constructs random gated network (RGN) and learns adversarially robust ensembles within the network. The method is inherently scalable, which can ideally instantiate numerous sub-models by sampling different paths within the RGN. By diversifying the vulnerabilities of different paths, the Ensemble-in-One method can efficiently obtain models with higher robustness, simultaneously reducing the overhead of model training and deployment. The individual model derived from the RGN shows much better robustness than previous ensemble training methods and achieves better trade-offs than adversarial training.

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

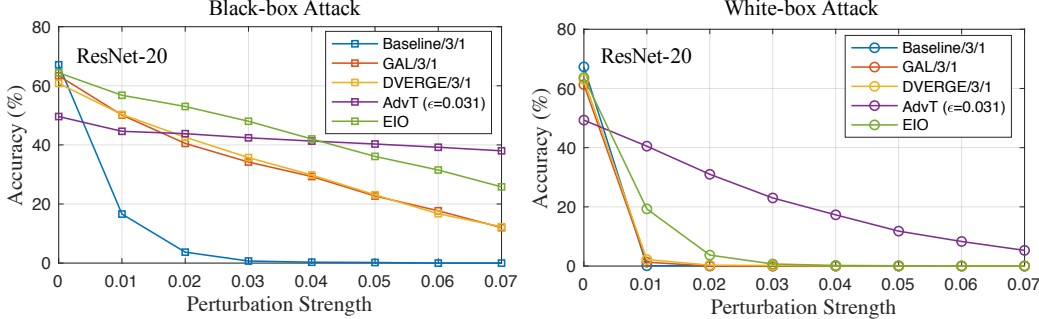

Figure 6: Contrasting the robustness of Ensemble-in-One with other methods on CIFAR-100 dataset. Because ensemble training brings significant clean accuracy enhancement on CIFAR-100 dataset (e.g. individual baseline 67.2% v.s. 3-sub-model ensemble baseline: 72.6%), we randomly sample an individual model from the ensemble for fair comparison. The number after the first slash represents the number of sub-models contained in the ensemble, and the number after the second slash represents that an individual model is sampled from each ensemble for evaluation. The *ADP* method is excluded because its training on CIFAR-100 fails to converge in our reproduction.

## A   APPENDIX

In this appendix, we provide some additional results to further discuss the advantages and disadvantages of our Ensemble-in-One method and other previous methods.

### A.1   ADVERSARIAL SETTINGS

We adopt strong attack methodologies and settings for evaluating the robustness of CNN models. For black-box transfer attacks, the involved attack methods include: (1) PGD with momentum and with three random starts (Madry et al., 2017); (2) M-DI$^2$-FGSM (Xie et al., 2019); and (3) SGM (Wu et al., 2020). The attacks are with different perturbation strength and the iterative steps are set to 100 with the step size of $\epsilon/5$. Besides the cross-entropy loss, we also apply the C&W loss to incorporate with the attacks. Therefore, there will be 3 (surrogate models) × 5 (attack methods, PGD with three random starts, M-DI$^2$-FGSM, and SGM) × 2 (losses) = 30 adversarial versions. For white-box attacks, we apply 50-step PGD with the step size of $\epsilon/5$ with five random starts. Both the black-box and white-box adversarial accuracy is reported in an *all-or-nothing* fashion: a sample is judged to be correctly classified only if its 30 (for black-box attack) or 5 (for white-box attack) adversarial versions are all corrected classified by the model. In default, we randomly sample 1000 instances from the test dataset for evaluation.

### A.2   CIFAR-100 EVALUATION

We further evaluate the methods on CIFAR-100 dataset which utilizes the ResNet-20 as a basic network. We apply the default hyper-parameter configurations for training the EIO network. The black-box setting utilizes 10 adversarial versions (2 surrogate models × 5 random starts applied in PGD) for transfer attacking the target model, and the white-box attack keeps the same setting as described before. As shown in Fig.6, EIO still significantly outperforms the other ensemble training methods whether under black-box or white-box attack scenarios. AdvT shows much better robustness while sacrificing significant clean accuracy. Overall, the results again suggest that EIO is a more effective approach for improving the adversarial robustness.

### A.3   TRAINING CURVE AND COST OF ENSEMBLE-IN-ONE

We investigate the training convergence of EIO. The training curves of baseline and EIO training are shown in Fig.7 (Left). Two training stages are involved in EIO training. One is the diversity training of the RGN, and the other is the fine-tuning of the extracted single path. EIO slightly deteriorates the convergence effectiveness than baseline training, while still obtaining competitive

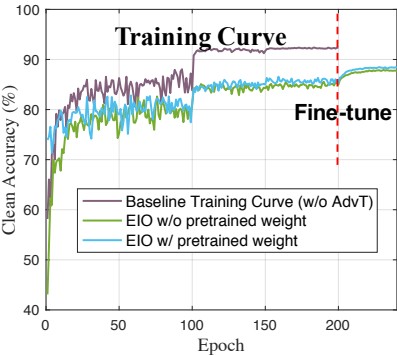 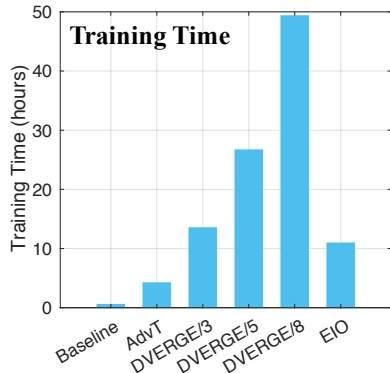

Figure 7: The training curves and time cost of Ensemble-in-One. The experiments are made on ResNet-20-based EIO training, with default hyper-parameters. Left: The clean validation accuracy throughout the training process. The fine-tuning epoch is extended to 40 epoch. As is observed, 20 epochs are adequate to help the derived model to fully converge. Right: The training time cost of Baseline, AdvT, DVERGE, and EIO, which are evaluated on an NVIDIA GeForce RTX 2080 Ti.

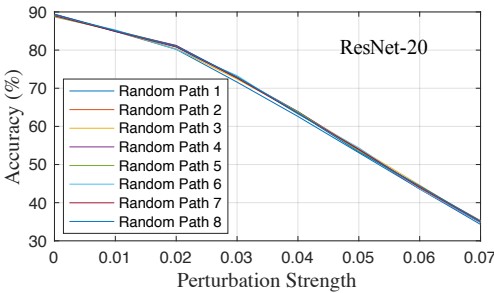 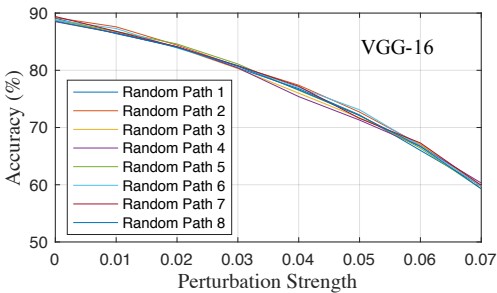

Figure 8: Investigation on the stability of derived models from the RGN. The experiments are made on CIFAR-10 dataset based on ResNet-20 and VGG-16 respectively. Black-box adversarial accuracy is evaluated.

clean accuracy after a slight fine-tuning. We also summarize the training time cost of different methods in Fig.7 (Right). Because we use a PGD-based data distillation for training and sample 3 paths per training iteration, the training time cost of an EIO network is approximately $2.5\times$ than AdvT. While the training time is substantially reduced compared to the DVERGE when scaling up the ensemble. As for the GPU memory and FLOPS cost, training EIO is closely equivalent to training one individual model because by controlling the gating, only one sub-model is involved at each training step. As is desired, our approach significantly reducing the training cost (both time and hardware cost), simultaneously obtaining better robustness than conventional ensemble training.

### A.4 MODEL DERIVATION STABILITY

In the deployment phase, an individual model (or several models) will be derived from the random gated network (RGN) and fine-tuned for a few epochs. Because the model is derived by randomly sampling a path in the RGN, it is important to ensure the stability of derived models. Hence, we randomly derive eight sub-models from a same RGN and test their performance and robustness. As can be observed from Fig.8, the sampled eight sub-models demonstrate almost the same robustness with very slight fluctuations on the adversarial accuracy against the black-box transfer attacks. Thus, the results can be seen as an evidence that the derivation is a stable process and the randomness would not bring high variance on the derived models.

