# OpenReview forum: "Ensemble-in-One: Learning Ensemble within Random Gated Networks for Enhanced Adversarial Robustness"
_ICLR.cc/2022/Conference — ICLR 2022 Submitted_

### Official Review · Reviewer_vVnY · 2021-10-28

**Correctness:** 3
**Technical Novelty And Significance:** 3
**Empirical Novelty And Significance:** 2
**Recommendation:** 5
**Confidence:** 3

**Main Review:**

1. It seems to me that this training is complicated and computationally expensive. The defender needs to consider (1) different data pairs (N^2) (2) all possible paths (although the authors choose p paths in each data batch, the training epoch should be much larger to cover possible paths) (3) generating different clean-label adversarial perturbations.

2. Some parts of the training process are quite confusing:  (1) My understanding of this training process is that it forces some paths to remember certain perturbation patterns. But how can the method guarantee that (a) the memorizations will not be rewritten by other perturbations?  (b) adversarial inputs will go into the corresponding paths in the inference stage? And what is the benefit of using such a method compared with forcing the neural network to remember all perturbations? (2) The objective here is to generate a perturbation that relies on the path and certain training data. In fact, Equation 2 and its variants can be used to generate clean-label attacks. However, its application here is not for defending such an attack. I don't understand why the authors choose to use equation 2 instead of using the conventional non-targeted adversarial attack. Also, if one considers $l$ to be the output layer, then basically, the adversarial generation in RGN reduces to a targeted attack. Of course, using the targeted attack in robust training is not plausible.

3. The inference phase of RGN searches a subnetwork to defend against attacks. This raises a question. Does the robustness come from its complicated training process, or does it come from the distillation? In fact, many works have explored the lottery ticket hypothesis in the standard training and robust training regimes, e.g., [1].

[1] Towards Practical Lottery Ticket Hypothesis for Adversarial Training

**Summary Of The Paper:**

This paper proposes a robust training and defending method RGN by applying control gates with binary status. During the training, the proposed method generates adversarial examples in a clean-label attack manner and mitigates the adversarial perturbation through training on another path. During the inference, RGN finds a subnetwork to defend against adversarial attacks.

**Summary Of The Review:**

Although applying control gates is an interesting idea, I have some concerns about the RGN method. I have listed my major concerns in the main review.

---

> ### Author Response · Authors · 2021-11-18
> **Responses on the training complexity, memorization, targeted attack, and the where the robustness come from. [Part 1/2]**
>
> We appreciate your questions on our work which pushes us to think deeper into our method. The belows are our responses.
>
> **Q1**. *It seems to me that this training is complicated and computationally expensive. The defender needs to consider (1) different data pairs (N^2) (2) all possible paths (although the authors choose p paths in each data batch, the training epoch should be much larger to cover possible paths) (3) generating different clean-label adversarial perturbations.*
>
> **[Training Complexity]**. Our method is more efficient than the conventional ensemble training method which usually needs an independent round-robin training process, as shown in Fig.7 in the paper. Belows are detailed analysis.
> -  Do we need to train every path in the super-network?
>
>     - We think it is not necessary. Because the sub-models within EIO super-network share a proportion of weights, their trainings are benefiting each other. Even a path has never been sampled during training, it still gains the ability because its operators are already well trained through other paths.
> -  Do we need to sample all possible data pairs?
>
>    - We also think it is not necessary. For the adversarial perturbation generation, the data pairs are selected randomly and it is also not required that all possible data pairs should appear. The sampling process is more likely a non-target adversarial perturbation generation because the paired data is absolutely random. In our implementation, we still treat an epoch as a round that go through the whole training dataset once.
> -  Do we need adversarial perturbations?
>
>     - Yes, the distillation process is needed. While as it is a most-used techniques in adversarial defensive approaches,  the time overhead is unavoidable. Actually, our approach is far more efficient than conventional ensemble training method, as presented in Fig.7 in the paper. For example, training an ensemble with 8 sub-models by DVERGE takes around 50 hours while EIO only takes near 13 hours.
>
> **Q2**. *Some parts of the training process are quite confusing: (1) My understanding of this training process is that it forces some paths to remember certain perturbation patterns. But how can the method guarantee that (a) the memorizations will not be rewritten by other perturbations? (b) adversarial inputs will go into the corresponding paths in the inference stage? And what is the benefit of using such a method compared with forcing the neural network to remember all perturbations? (2) The objective here is to generate a perturbation that relies on the path and certain training data. In fact, Equation 2 and its variants can be used to generate clean-label attacks. However, its application here is not for defending such an attack. I don't understand why the authors choose to use equation 2 instead of using the conventional non-targeted adversarial attack. Also, if one considers l to be the output layer, then basically, the adversarial generation in RGN reduces to a targeted attack. Of course, using the targeted attack in robust training is not plausible.*
>
> **[Memorization]**. There are two types of possible forgetting phenomena: 1) At the level of adversarial examples/perturbation: Early studies[1] indeed keep a large pool of black-box adversarial examples and force the NN to remember all these perturbations. But recent studies[2, 3] do not adopt this manner to avoid overfitting into these specific perturbations. And we follow this practice to jointly generate adversarial perturbations and train the model. 2) At the level of different models/paths: As training a path might overwrite the parameters of other paths, there indeed exists some sort of forgetting phenomenon in the parameter-sharing training process. This phenomenon is also studied in the NAS literature [4,5], and their developed techniques can be effortlessly incorporated (e.g., temporal weights ensemble) with our approach. Moreover, we do not anticipate the forgetting phenomenon caused by inter-model overwrting to be that severe in our case (compared to NAS), since all  paths have the same architecture.
>
> [1] Liao, Fangzhou, et al. "Defense against adversarial attacks using high-level representation guided denoiser." Proceedings of the IEEE Conference on Computer Vision and Pattern Recognition. 2018.
>
> [2] Yang, Huanrui, et al. "DVERGE: diversifying vulnerabilities for enhanced robust generation of ensembles." arXiv preprint arXiv:2009.14720 (2020).
>
> [3] Zhang, Hongyang, et al. "Theoretically principled trade-off between robustness and accuracy." International Conference on Machine Learning. PMLR, 2019.
>
> [4] Yassine Benyahia, Kaicheng Yu, Kamil Bennani Smires, Martin Jaggi, Anthony C Davison,Mathieu Salzmann, and Claudiu Musat. Overcoming multi-model forgetting. ICML 19.
>
> [5] Ronghao Guo, Chen Lin, Chuming Li, Keyu Tian, Ming Sun, Lu Sheng, and Junjie Yan. Powering one-shot topological nas with stabilized share-parameter proxy. ECCV 20.

---

> > ### Author Response · Authors · 2021-11-18
> > **Responses on the training complexity, memorization, targeted attack, and the where the robustness come from. [Part 2/2]**
> >
> > **[About Non-targeted Attack]** We agree that current distillation uses a form of "targeted" perturbation generation: The perturbations are crafted to make the target data have similar features as the source data. However, the target of the overall training process is still "non-targeted", because the data pair selection is absolutely random, and the optimization objective is an expectation w.r.t. this random selection of all other classes.
> >
> > **Q3**. *The inference phase of RGN searches a subnetwork to defend against attacks. This raises a question. Does the robustness come from its complicated training process, or does it come from the distillation? In fact, many works have explored the lottery ticket hypothesis in the standard training and robust training regimes, e.g., [1].*
> >
> > **[Where does the improvement come from]** We would say all the techniques we introduced are helpful. As our experiments show, applying adversarial distillation, random gating, and mutual learning among different sub-models in the model training can obtain the following black-box adversarial accuracy:
> >
> > |method                                  | 0.00 | 0.01 | 0.02 | 0.03 | 0.04 | 0.05 | 0.06 | 0.07|
> > | -- | -- | -- | -- | -- | -- | -- | -- | -- |
> > baseline                                 | 91.2  | 0.1 | 0 | 0 | 0 | 0 | 0 | 0
> > distillation                              | 84.7 | 82.0 | 76.7 | 69.8 | 60.6 | 49.7 | 38.4 | 29.1
> > random gating + self distillation | 87.3 | 83.2 | 77.3 | 70.3 | 62.4 | 51.3 | 41.1 | 32.5
> > EIO (random gating + mutual distillation)  | 89.3 | 84.9 | 80.2 | 71.6 | 62.6 | 53.1 | 43.6 | 34.3
> >
> >
> > As can be seen, compared to the baseline, the distillation gains higher robustness while losing significant clean (natural) accuracy. With random gating, both the clean accuracy and robustness can be enhanced. Applying the mutual learning of different sub-models can further improve the performance.

---

### Official Review · Reviewer_7WHx · 2021-11-01

**Correctness:** 3
**Technical Novelty And Significance:** 2
**Empirical Novelty And Significance:** 2
**Recommendation:** 5
**Confidence:** 4

**Main Review:**

Strengths:
1. The idea of using random gated network (RGN) to make the ensemble scalable is very interesting.
2. Strong empirical results when compared with SOTAs.

Weakness:
1. Instead of just presenting the numerical results, the authors could try to provide some theoretical analysis on the proposed methods. A good example could be the ICLR 2018 paper: Ensemble Adversarial Training: Attacks and Defenses
2. The concept of the random gated network is similar to the work of stochastic path network (``Deep Networks with Stochastic Depth").
3. The authors discussed two ways to deploy the final model in the testing phase. Why not consider the case where we just aggregrating the results, possibly by average  pooling, from different paths at each layer ?

**Summary Of The Paper:**

This paper uses ensemble learning to improve the adversarial robustness. The authors introduce the concept of ``Ensemble-in-One",  in which a simple but effective method called random gated network (RGN) is ultilized to enlarge the ensemble size in a very effective manner.

**Summary Of The Review:**

The idea is quite interesting and the results are good. However, the current version lacks the theoretical analysis and other key empirical comparisons as mentioned above.

---

> ### Author Response · Authors · 2021-11-18
> **Responses on theoretical analysis and layer aggregation**
>
> We appreciate your insightful comments on our paper and provide practical suggestions.  Belows are our responses.
>
> **Q1**. *Instead of just presenting the numerical results, the authors could try to provide some theoretical analysis on the proposed methods. A good example could be the ICLR 2018 paper: Ensemble Adversarial Training: Attacks and Defenses.*
>
> **[Theoretical Analysis]**. Thanks for the suggestion and the related paper recommendation. We are trying to find the theoretical support for our approach. While ensemble-in-one is intuitively designed and empirically demonstrated, we will seek for the behind working mechanism in the following revision.
>
> **Q2**. *The concept of the random gated network is similar to the work of stochastic path network (``Deep Networks with Stochastic Depth").*
>
> **[Differences]**. We think the motivation and goal of the mentioned work are both different with EIO. The stochastic depth introduced in the work aims at solving the gradient vanishing problem in deep networks, which enables random drop of a subset of layers and bypass them with identity function. Our method random chooses one operator in each augmented layer to construct a very large number of sub-models.
>
> **Q3**. *The authors discussed two ways to deploy the final model in the testing phase. Why not consider the case where we just aggregating the results, possibly by average pooling, from different paths at each layer ?*
>
> **[Layer Aggregation]**. One main motivation of designing Ensemble-in-One is that the conventional way to enlarge the ensemble will incur significantly larger computational and storage cost. Therefore, EIO is designed to efficiently scale up an ensemble by setting the random gating mechanism within a large super-network. In our setting, the both two deployment methods will not increase the inference overhead because for an input, whichever path is chosen, the computation graph is exactly the same as the original network. While if aggregating the results from different paths at each layer, it incurs more computation as we must compute multiple outputs on each layer. Moreover, since the aggregation was absent during training, the inference accuracy will significantly drop. Nevertheless, aggregating the features on each layer may also be a good idea. We will try that in future work.

---

> ### Comment · Reviewer_7WHx · 2021-11-25
> **response for the rebuttal**
>
> I will keep my initial rating as my previous concerns are not properly addressed.   Apart from the theoretical part, which is not provided in the current version, the proposed method shares a certain level of similarity with the stochastic path network. In addition, the authors should try to investigate more advanced aggregation approaches.

---

### Official Review · Reviewer_a3La · 2021-11-02

**Correctness:** 2
**Technical Novelty And Significance:** 3
**Empirical Novelty And Significance:** 3
**Recommendation:** 6
**Confidence:** 4

**Main Review:**

Pros:
1.The paper is well written and easy to read.
2.The idea is interesting and clearly explained.
3.Experimental setup is comprehensive and appropriate ablation studies have been performed.

Cons:
1.Adversarial training (AT) seems a better option. As shown in Figure 5&6, the AT can easily beat EIO as perturbation strength increase from 0.01 under the White-box setting. Considering the AT only takes less than half the training time of EIO (Figure 7), it makes EIO less attractive. It seems that EIO can be easily combined with AT together. I would suggest that the author conducts the experiment to check the performance of the combination of EIO and AT.


2.Lack of interpretation of results. Since the DVERGE and EIO share the same vulnerability diversification training, why EIO can outperform DVERGE significantly? Is it because EIO can impede more adversarial transferability among sub-models within the ensemble? In Figure 6, why do you only take one sub-model in the ensemble to evaluate the performance of GAL and DEVRGE? Is that the reason why GAL and DEVRGE completely fail to defend against White-box attacks?

3.The major contribution of the paper is to construct ensemble by repeating the original layer by n times. However, the increase of this key parameter n (from 2 to 3) will lead to performance degradation (Figure 3), which does not match the intuition of the paper.  Given enough training time, will the model trained with n=3 beat the model trained with n=2?

Minor:
1.	Table 1 is not clear, what does the number mean before and after the slash, is that accuracy against White-box/Black-box attacks?
2.	There are typos in legend of Figure 4 (Baseline.5 and DVERGE.8).


**Summary Of The Paper:**

This paper proposed a new way to generate an ensemble of networks against adversarial attacks. Different from other methods, which train different sub-models, the proposed method repeats convolution layers multiple times and controls them with random gates.  The experiment demonstrates that it outperforms other ensemble training with a smaller computational overhead.

**Summary Of The Review:**

Overall, I think this paper is marginally below the acceptance threshold.  I like the idea of the ensemble with random gates in deep neural networks. Although this method can beat other ensemble methods, it is less attractive than basic adversarial training since the latter takes less training time and is flexible to be tuned with different perturbation strengths. If my concerns are addressed, I would like to raise the score.

---

> ### Author Response · Authors · 2021-11-18
> **Responses on AT, result interpretation, and the ensemble augmentation [Part 1/2]**
>
> We appreciate your thorough review of our paper and conclude the pros and cons. We add experiments following your suggestions. Belows are our responses.
>
> **Q1**. *Adversarial training (AT) seems a better option. As shown in Figure 5&6, the AT can easily beat EIO as perturbation strength increase from 0.01 under the White-box setting. Considering the AT only takes less than half the training time of EIO (Figure 7), it makes EIO less attractive. It seems that EIO can be easily combined with AT together. I would suggest that the author conducts the experiment to check the performance of the combination of EIO and AT.*
>
> **[Combination with AT]** First of all, Fig.5 might be a little misleading. Compared with AT, EIO focuses on the black-box robustness and achieving a better trade-off of the clean and adversarial accuracy. As shown in Fig.5, when aligning the clean accuracy of AT and EIO, EIO significantly outperforms AT under black-box attack settings. Combination of EIO and AT is an interesting idea, as has also been tried in DVERGE. Following your suggestion, we try three ways of incorporating AT with EIO. They are: (1) Replace the feature distillation with a AT process + random gating + mutual learning of different paths; (2) Using distilled data for mutual learning + AT for self learning (self learning means the adversarial perturbation will be trained on the path where they are generated); and (3) AT + random gating + self learning.
>
> | Method | clean | 0.01 | 0.02 | 0.03 | 0.04 | 0.05 | 0.06 | 0.07 |
> | --- | --- | --- | --- | --- | --- | --- | --- | --- |
> |(1) | fail | - | - | - | - | - | - | - |
> |(2) | 82.6 | 80.1 | 77.9 | 74.5 | 69.8 | 65.7 | 60.7 | 53.3|
> |(3) | fail | - | - | - | - | - | - | - |
>
> However, we find that under the settings (1) and (3), the training will not converge well. Under setting (2), we do not obtain much better performance than vanilla EIO. How to combine ensemble learning techniques (like EIO) and AT techniques is worth studying in the future work, we will continue to make more attempts.
>
> **Q2**. *Lack of interpretation of results. Since the DVERGE and EIO share the same vulnerability diversification training, why EIO can outperform DVERGE significantly? Is it because EIO can impede more adversarial transferability among sub-models within the ensemble? In Figure 6, why do you only take one sub-model in the ensemble to evaluate the performance of GAL and DEVRGE? Is that the reason why GAL and DEVRGE completely fail to defend against White-box attacks?*
>
> **[Interpretation of Results]** We analyze that EIO's improvements w.r.t. DVERGE come from the following aspects. On one hand, EIO constructs a much larger ensemble (in a efficient way) which will provide richer vulnerability diversity. Therefore, each sub-model can learn to avoid the vulnerability of more models. On the other hand, instead of only being a technique to trade off the performance for efficiency, sharing the parameters between different models might also bring performance improvements, since every module can learn from diverse forward and backward paths to reduce the co-adaptation between different layers (Similar insights or techniques can be found in [1][2]). We will discuss these interpretation in our paper more formally and provide verification results.
>
> **[CIFAR-100 Experiments]** We take only one sub-model from the ensemble of Baseline, GAL, and DVERGE for fair comparison when considering the same computation budget, because in CIFAR-100 dataset, the accuracy is significantly boosted when aggregating multiple sub-models' predictions. The clean accuracy of the ensemble is much higher than an individual model (e.g. in baseline 74.8% v.s. 67.2%). Moreover, this is not the main reason why GAL and DEVRGE fail to defend against white-box attacks. We add the experiments which include the whole ensembles as below. As can be seen, EIO still demonstrate  better robust accuracy than these ensemble  methods.
>
> |Method | clean | 0.01 | 0.02 | 0.03 | 0.04 | 0.05 | 0.06 | 0.07|
> | -- | -- | -- | -- | -- | -- | -- | -- | -- |
> baseline/3/1 | 67.2	| 0.1 |	0.1 |	0	| 0 |	0 | 0| 	0|
> baseline/3  |74.8 | 0.1 | 0| 0| 0| 0| 0| 0|
> dverge/3/1  | 63	| 2.2|	0.3|	0.2|	0.1|	0.1|	0|	0|
> dverge/3   | 69.1| 12.2| 2.7| 1.1| 0.4| 0.3| 0.1| 0|
> gal/3/1   |61.3|	1.3|	0	|0	|0	|0	|0	|0|
> gal/3            | 67.6 |10.2| 1.8| 0.3| 0.2| 0| 0| 0|
> EIO/1         |  63.7| 19.3| 3.7| 0.7| 0.2| 0.1| 0 0| 0|
>
>
>
>
>
>
> [1] Baldi, Pierre, and Peter J. Sadowski. "Understanding dropout." Advances in neural information processing systems 26 (2013): 2814-2822.
>
> [2] Szegedy, Christian, et al. "Going deeper with convolutions." Proceedings of the IEEE conference on computer vision and pattern recognition. 2015.  (Auxiliary tower/head for training)

---

> > ### Author Response · Authors · 2021-11-18
> > **Responses on AT, result interpretation, and the ensemble augmentation [Part 2/2]**
> >
> > **Q3**. *The major contribution of the paper is to construct ensemble by repeating the original layer by n times. However, the increase of this key parameter n (from 2 to 3) will lead to performance degradation (Figure 3), which does not match the intuition of the paper. Given enough training time, will the model trained with n=3 beat the model trained with n=2?*
> >
> > **[n=2 versus n=3]** Intuitively, augmenting more operators enable a larger ensemble construction, and should achieve better results, given enough training budgets. Following your suggestion, we prolong the training epochs from 200 to 400 when n=3, and we obtain a robustness improvement.  We make several ablation studies as follows. S1 means only augmenting the first stage in ResNet-20.
> >
> > |Aug  |       0.00   | 0.01 |  0.02 | 0.03 |  0.04 |  0.05 | 0.06 |  0.07|
> > | -- | -- | -- | -- | -- | -- | -- | -- | -- |
> > |None   | 84.7 | 82.0 | 76.7 | 69.8 | 60.6 | 49.7 | 38.4 | 29.1 |
> > |S1(n=2) | 87.1 | 82.6 | 77.0 | 71.8 | 62.0 | 53.3 | 44.2 | 33.3|
> > |n=2 |  89.3 | 84.9 | 80.2 | 71.6 | 62.6 | 53.1 | 43.6 | 34.3|
> > |n=3 |  87.5 | 82.4 | 78.1 | 73.8 | 67.2 | 59.1 | 49.2 | 39.6|
> >
> > We can see the trend that the robustness improves with a stronger augmentation. While as we mentioned in the discussion section, the augmentation way for constructing EIO is still worth further exploration.
> >
> > **Minor**.
> > - *Table 1 is not clear, what does the number mean before and after the slash, is that accuracy against White-box/Black-box attacks?*
> > - *There are typos in legend of Figure 4 (Baseline.5 and DVERGE.8).*
> >
> > **[Typos]** Thanks for your careful catch. We will revise the typo in the manuscript and add an explanation in Table 1.

---

> > > ### Comment · Reviewer_a3La · 2021-11-29
> > > **Part of my concerns has been addressed. The rating has been adjusted.**
> > >
> > > Thanks for the detailed response, it addressed part of my concerns in Q2 & Q3. I would increase my rate from 5 to 6.
> > > Overall, I think the idea is interesting, it shows some improvements over other ensemble methods. However, the major issue is that it does not show a significant advantage over vanilla adversarial training.  The performance of EIO and AT is close, while the latter is can be easily tuned with different perturbation strengths and only takes half of the training time. These will make  EIO less attractive to the community.

---

### Official Review · Reviewer_wUHR · 2021-11-03

**Correctness:** 3
**Technical Novelty And Significance:** 2
**Empirical Novelty And Significance:** 3
**Recommendation:** 5
**Confidence:** 3

**Main Review:**

1. The random gated block enable the efficient/scalable model ensemble but the introduced randomness may greatly decrease the model capacity. It would be great if ImageNet dataset can be tested to prove algorithm can fit highly complex data distribution.
2. The random gated method looks very similar to applying dropout not only during the model training phase but also the model inference phase. It would be good to clearly address the key difference between random gating and dropout. It would be expected that some similar theoretical discussions like the following paper can be included in the draft.
(1) Gal, Yarin, and Zoubin Ghahramani. "Dropout as a bayesian approximation: Representing model uncertainty in deep learning." In international conference on machine learning, pp. 1050-1059. PMLR, 2016.


**Summary Of The Paper:**

This paper proposes a scalable ensemble training with random gated block to enhance the model adversarial robustness.

**Summary Of The Review:**

The model capacity can be a big concern and the technical depth is somehow limited.

---

> ### Author Response · Authors · 2021-11-18
> **Responses on the model capacity and differences with dropout**
>
> We appreciate your valuable comments and suggestions on our work. We conclude our responses as follows.
>
> **Q1**: *The random gated block enable the efficient/scalable model ensemble but the introduced randomness may greatly decrease the model capacity.  It would be great if ImageNet dataset can be tested to prove algorithm can fit highly complex data distribution.*
>
> **[Model Capacity]** We agree that inference-time randomness may  affect the model capacity. But our method only introduces randomness in the training time, and  the derived final model is a fixed subnetwork of the training-time supernet, without inference-time randomness. In another word, given an original network, although we transform it into a random gated network for better adversarial training,  the final derived model is exactly the same as the original network (i.e., has the same model capacity).
>
>
> Due to the limited rebuttal period, we are sorry that we are unable to deliver the results on ImageNet at this moment (as will take several weeks). We will add such more complicated experiments in the following revision. While we have demonstrated the effectiveness on a more complicated dataset CIFAR-100, and we observe consistent robustness improvement compared to other methods, as can be seen in Fig. 6 in the paper. We also have made experiments on a larger network WideResNet-34 with CIFAR-10 dataset, the black-box adversarial accuracy is shown as below:
>
> |Method|  clean | 0.01 | 0.02 | 0.03 | 0.04 | 0.05 | 0.06 | 0.07 |
> | --------- | ------ | ------ | ---- | --- | ---| --- | --- |--- |
> |Baseline |       94.5 | 20.0 | 0.6 |  0.0 | 0.0  | 0.0 |  0.0 | 0.0 |
> |TRADES[1] |   84.6 | 83.3 | 82.2 | 79.5 | 78.0 | 76.0 | 72.9 | 69.9 |
> |EIO |           91.2 | 89.7|  87.8|  85.2|  81.6|  77.0|  73.6 | 68.2|
>
> Compared to the  recent work TRADES, EIO achieves both better clean accuracy and black-box adversarial accuracy. Moreover, EIO still maintains comparable clean accuracy as the vanilla baseline.
>
> **Q2**: *The random gated method looks very similar to applying dropout not only during the model training phase but also the model inference phase. It would be good to clearly address the key difference between random gating and dropout. It would be expected that some similar theoretical discussions like the following paper can be included in the draft. (1) Gal, Yarin, and Zoubin Ghahramani. "Dropout as a bayesian approximation: Representing model uncertainty in deep learning." In international conference on machine learning, pp. 1050-1059. PMLR, 2016.*
>
> **[Differences with Dropout]** Yes, the random gated training technique can be seem as dropping out paths (drop-path) from the supernet during training. And we will add the following discussion to clarify our differences with dropout: Actually, the aim of our work is vastly different from this vanilla usage of dropout. Our aim is to construct a large amount of models to learn from each other, but solely because we share their parameters for efficiency, it can be seen as dropping out other paths in a supernet when forward a model.
>
> We appreciate the reviewer to recommend the work [2]. It gives out an Bayesian interpretation of why dropout helps improve the uncertainty modeling of a model. But this vanilla type of dropout technique plays a different role as our method. Metaphorically, the vanilla  technique is dropping "inside" a network at the parameter-level, thus can be viewed as a prior on the parameters and improve the uncertainty modeling of the final parameters. Whereas our training technique can be seen as dropping "outside" a network, thus do not work in the same way as dropping "inside" a network. Look at it another way (from the perspective of ensemble techniques), besides the differences in dropout granularity, vanilla dropout can be seen as training and testing with a very large ensemble of networks, whereas the testing / usage in our method does not use the training-time ensemble, but a fixed derived sub-model instead.
>
> **[Theoretical Discussion]** Thanks for the suggestion. The design of Ensemble-in-One was driven by the intuition and empirical analysis, and indeed lack of theoretical support. We will try to find the behind working mechanism in the following revision.
>
> [1] Hongyang Zhang, Yaodong Yu, Jiantao Jiao, Eric Xing, Laurent El Ghaoui, and Michael Jordan. Theoretically principled trade-off between robustness and accuracy. In International Conference on Machine Learning, pp. 7472–7482. PMLR, 2019.
>
> [2] Gal, Yarin, and Zoubin Ghahramani. "Dropout as a bayesian approximation: Representing model uncertainty in deep learning." In international conference on machine learning, pp. 1050-1059. PMLR, 2016.

---

### Decision · Program_Chairs · 2022-01-20

**Decision:**

Reject

**Comment:**

The paper proposes a stochastic network, named Ensebmle-in-One (EIO), to increase adversarial robustness. EIO replaces the layers in a given architecture by so called random gated blocks (RGBs) in which a random gate switches between multiple copies of the original layers. By sampling from the random gates different subnetworks can be sampled which can be arranged to form an ensemble. During training non-robust feature distillation (as proposed in previous work) between models is applied. For inference in the experiments a single subnetwork is sampled, and the robustness of that subnetwork is compared against several ensemble methods and adversarial training.

One reviewer was worried about model capacity and recommended to perform experiments on image net to demonstrate scalability to large datasets. In turn, authors added experiments on CIFAR-100 during rebuttal period. Another critique was that the model does not show a significant advantage over vanilla adversarial training (AT), which can easily tuned with different perturbation strengths and only takes half of the training time. While other ensemble techniques, like DVERGE, can be combined with AT to improve their robustness, combing EIO with AT does not lead to improvements as shown by experiments performed during the rebuttal period. Two reviewers stated that adding a theoretical analysis will improve the paper. Another suggestion for improving the paper was to add a comparison to stochastic path networks, which is related work, and to investigate model performance when results from several sub-networks are aggregated.

Overall, the paper can not be accepted in its current state, but I would recommend the authors to continue the direction of work and to incorporate reviewers suggestions in a future version of the manuscript.